# Farmers' Beliefs and Concerns about Climate Change: An Assessment from Southern Saudi Arabia

**Bader Alhafi Alotaibi** [1,*] , **Hazem S. Kassem** [1,2] , **Roshan K. Nayak** [3] **and Muhammad Muddassir** [1]

1   Department of Agricultural Extension and Rural Society, King Saud University, Riyadh 11451, Saudi Arabia; hskassem@ksu.edu.sa (H.S.K.); mrajpoot@ksu.edu.sa (M.M.)
2   Department of Agricultural Extension and Rural Society, Mansoura University, Mansoura 35516, Egypt
3   California 4-H Foundation, University of California Agricultural and Natural Resources, Davis, CA 95616, USA; rknayak@ucanr.edu
*   Correspondence: balhafi@ksu.edu.sa; Tel.: +966-504240201

**Abstract:** Climate change constitutes a major threat to agricultural production, food security, and natural resource management. Saudi Arabia is particularly susceptible to increasing temperatures and extreme climatic events, such as arid weather and drought. The purpose of this study is to assess farmers' beliefs and concerns as regards climate change. Extensive interviews were conducted with 164 farmers in the Jazan region. Results revealed that 89.6% of the farmers believed that climate change is due to human activities and 93.3% believed that it is because of natural change. Seventy-five percent of the farmers were concerned about insects and 73% about the prevalence of weeds on their farms. Findings of cluster analysis revealed that farmers who are more likely to believe in climate change are more in agreement with the role of extension services in capacity building. Farmers' beliefs about climate change were significantly influenced by membership of agricultural cooperatives, access to loans, use of extension services, age, farm size, and level of soil fertility. Access to loans was the only significant factor to explain the differences in farmers' concerns. These results suggest the need for capacity-building activities targeted at improving farmers' adaptability to manage climate variability.

**Keywords:** beliefs; concern; climate change; farmers; capacity-building; Saudi Arabia

## 1. Introduction

In recent years, many counties have experienced various adverse effects of climate change. This issue gained more attention in 2015 when the United Nations adopted Goal 13, "take urgent action to combat climate change and its impact", as one of its Sustainable Development Goals (SDGs) [1]. Consequently, governments around the world have begun to implement multiple initiatives and national plans to address climate change, not only to mitigate the adverse consequences associated with the issue and adapt to it, but also to cover the other SDGs [2]. According to Iheke and Agodike [3], climate and environmental change processes lead to changes in atmosphere, water resources, soil, and land surface. The rapid changes associated with climate change are predicted to have negative effects on food security and sustainable livelihoods [4].

Climate change affects the agricultural sector in a number of ways. Changes in weather, including fluctuations in temperature and precipitation, and extreme weather events, are already impacting crop yields, the availability of water for irrigation, and livestock productivity. Weather and climate conditions also affect the processing of agricultural products, as well as transportation and storage conditions [5]. Furthermore, farming livelihoods are negatively affected by the adverse impacts

of climate variability and change due to dependence on climate-sensitive, natural resource-based economic activities [6].

In Saudi Arabia, the rural population and farming systems are highly reliant on favorable climatic conditions. Due to the country's arid climate, managing the sustainability of water resources is extremely important. Therefore, high variability of rainfall and climate change negatively affects farming families living in rural areas [7]. Several studies have produced modeling projections that show the potential for adverse consequences due to climate change in Saudi Arabia. Chowdhury and Al-Zahrani [8] forecasted a temperature increase from 2.1 to 4.1 °C in the northern region of Saudi Arabia and from 3 to 4.1 °C in the northwest by 2050. Another study revealed that Saudi Arabia's average temperature could increase by 6 °C by 2100. Both scenarios could dramatically increase the demand for agricultural irrigation water and lead to a reduction in fruit and other crops yields by between 5% and 25%. Such increases in temperature would escalate evapotranspiration by 10.3–27.4% and alter the rainfall pattern for the northern region of the country. Moreover, rainfall in the western region could increase by 109.7–130.4 mm/year by 2050 [9]. In the same vein, Tarawneh and Chowdhury [10] presented the rainfall and temperature patterns in the central north and southwest regions for the periods 2025–2044, 2045–2064, and 2065–2084 compared with the average values from the reference period, 1986–2005. Results indicated increases in temperature of 0.8–1.6 °C, 0.9–2.7 °C, and 0.7–4.1 °C during 2025–2044, 2045–2064, and 2065–2084, respectively. On the contrary, variable rainfall patterns were estimated for most regions during the same periods. Such changes could increase uncertainty in the development of sustainable water resource management strategies.

Understanding farmers' beliefs and concerns regarding the associated impacts of climate change on agriculture is a significant step toward climate change adaptation [11]. Beliefs can be formed based on farmers' experiences associated with climate change [12]. According to Al-Mutairi et al. [13], belief is an important factor in the adoption of climate change adaptation practices. In other words, if farmers do not perceive climate change as a threat to their livelihoods or if they do not believe that it is occurring, they are unlikely to act to adapt to or mitigate climate change [14]. A number of studies indicate that farmers' beliefs and motivations are positively influenced by the interventions of extension services in addressing climate change issues [2,15–17]. Extension services can help to create awareness, build resilience capacity among vulnerable farmers, and broker agreement between stakeholders on climate change issues [18].

According to the 2030 strategic plan of the Ministry of Environment, Water, and Agriculture (MEWA) [19], there is a weak understanding regarding the impact of climate change on people's livelihoods and their needs in terms of capacity building of areas to adapt with climate change in Saudi Arabia. However, most research conducted on climate change in Saudi Arabia has followed a top-down approach to predict the consequences of climate change on agricultural productivity and water resources. Only Al-Mutairi et al. [13] have taken a bottom-up approach, exploring how people in the Tabouk region in North Saudi Arabia conceptualize climate-related risk. Therefore, a research gap exists with regard to understanding how farmers with different characteristics perceive climate change. This research sought to fill that gap. Section one presents the research and analysis methods used to analyze farmers' beliefs and concerns regarding climate change. Second, farmers' beliefs and concerns about climate change are identified, and then to determine if these differ across demographic variables. Finally, farmers' perceptions of the role of extension services in capacity building to adapt with climate change are determined. The results of this study will contribute to closing the knowledge gap by elucidating the specific variables that influence farmers' beliefs and concerns. This will help in the development of suitable extension programs and awareness campaigns for rural Saudi Arabia. Furthermore, the results will aid the implementation of the needed transformational shifts in the farming systems and adoption of new technologies. This transformation will not only help in combating climatic changes but also in the achievement of the SDGs by 2030.

## 2. Methodology

### 2.1. Description of the Study Area

Jazan region is located in the southwest of Saudi Arabia along the Red Sea coast. The region is approximately 11,671 km$^2$ in area and covers 300 km of the Southern Red Sea coast. Administratively, Jazan region consists of 16 governorates; Al-Darb, Al-Reath, Beash, Haroob, Al-Daer, Savya, Al-Idabi, Faifa, Damad, Al-Aridah, Abu Arish, Jazan, Al-Harth, Ahad- Al-Musrarihah, Samttah, and Al Twal. The region lies between 42.33° E and 16.53° N, as shown in Figure 1, and is characterized by fertile loamy soil [20]. Despite the region representing only 0.7% of the total area of the country, it is one of the richest agricultural regions and contains approximately 8% of Saudi Arabia's farms [21]. Most farms in Jazan region are small, averaging 2–4 ha in size. It is considered the capital of Saudi Arabia's mango production, with approximately 750,000 trees producing 35,000 t annually. The region also produces sesame, millet, maize, okra, and tomatoes. Annual rainfall varies from year to year, with an average of 55–150 mm; a large amount of rainfall is observed between October and January. In summer, the temperature normally ranges from 31 to 35 °C, while the winter range is from 25 to 28 °C [22]. Water is sourced from fresh ground water, rain, and flash floods. Drip irrigation has been adopted by the majority of farmers using rainwater harvesting techniques.

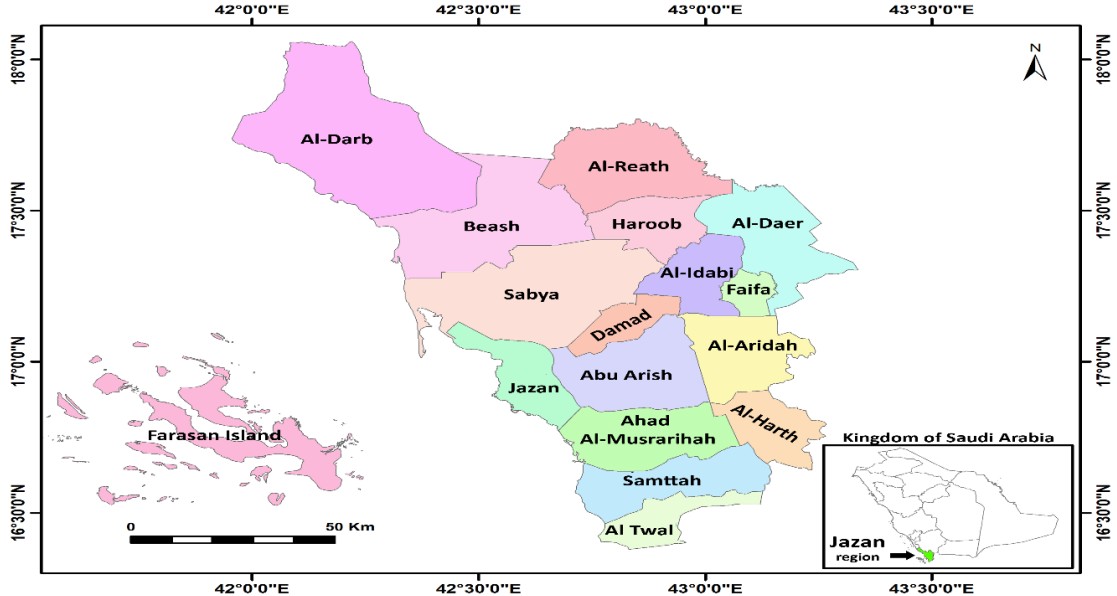

**Figure 1.** Map of the study area.

According to a few studies [9,23–25], Jazan region is already experiencing the manifestations of climate change in the form of droughts, floods, rising summer temperature, altered rainfall patterns, strong winds, land degradation in coastal areas, changes in weed species and distribution, and increased pest and disease pressures.

### 2.2. Research Design

A cross-sectional survey was designed to collect data from Jazan farmers. The data were collected during the period January–March 2019. Abu Arish governorate in central Jazan was randomly selected for data collection. The survey was developed and validated by a group of experts including extension agents. The data were completed by visiting farmers at their farms, as well as by meeting the farmers at extension centers. A total of 200 farmers invited to participate in the study; 164 completed the paper-based questionnaires, resulting in an 82% response rate. Prior to data collection, the purpose of the research project was explained to the farmers and they were assured that the information gathered

would only be used for academic purposes. Moreover, they were informed that their participation was not compulsory [26].

### 2.3. Instrument

The questionnaire was divided into four parts. The first part comprised questions concerning socio-economic information, including age, educational level, farming experience, types of farming activities, access to extension services, membership of agricultural cooperatives, farm size, access to loans, and level of soil fertility. The second section explored farmers' beliefs about climate change; these beliefs were measured using five statements, which were adopted and modified based on a study by Arbuckle et al. [14]. Respondents were asked to indicate their level of agreement or disagreement on a 5-point Likert-type scale, where 1 = Strongly Disagree, 2 = Disagree, 3 = Neither, 4 = Agree, 5 = Strongly Agree. The summed scores for farmers' beliefs were used to determine the level of farmers' beliefs regarding climate change. The total score of each respondent ranged from 5 to 25 and was converted to a percentage. Farmers' beliefs are classified as high beliefs—more than 75%; moderate—between 50 and 75%; and low beliefs—less than 50%.

The third part of the survey measured farmers' concerns about adverse impacts associated with climatic changes. Eight items were developed to measure farmers' concerns based on the adverse impacts addressed in previous research conducted in the study area. Farmers were asked to indicate their concern on a 4-point Likert-type scale, where 1 = Not Concerned and 4 = Very Concerned. The fourth part of the survey explored farmers' perceptions of the role of agricultural extension in capacity building with regard to climate change issues. Ten statements were adopted and modified as regards different areas of capacity building that agricultural extension can provide [27]. Farmers were asked to rate the level of importance of these statements on a 5-point Likert-type scale, where 1 = Strongly disagree and 5 = Strongly agree. Total concern scores were calculated out of 100 and divided into three categories according to the following range: Low ≤ 50%, Moderate = 50–75%, and High ≥ 75%.

### 2.4. Data Analysis

Descriptive statistics, using frequencies, percentages, means, and standard deviations, were used to address the research objectives. Analysis of variance (ANOVA) was employed to determine if farmers' beliefs and concerns regarding climate change differed across the demographic variables of age, level of education, type of crop, farm size, and level of soil fertility. A *t*-test was used for the variables: membership of agricultural cooperatives, access to loans, and access to extension services. The significance of differences was tested based on the summed scores of farmers' beliefs and concerns.

Principle axis factoring (PAF) was performed to extract factors from the statements related to climate change that asked farmers to rate their level of agreement. The PAF analysis used correlation matrix and varimax rotation to extract factors based on eigenvalues, and the results produced two factors, explaining 49.53% of the total variance. The items representing Factors 1 and 2 were based on their factor loading values. Items grouped under Factor 2 were determined to further analyze group respondents based on their beliefs regarding the occurrence of climate change. The K-mean cluster analysis of Factor 2 items produced two clusters of respondents. An independent sample *t*-test of those clusters was conducted for the two items to examine the characteristics of respondents in the two groups. The two clusters were used to study farmers' perceptions of the role of extension agents in building climate change adaptation capacity.

## 3. Results and Discussion

### 3.1. Farmers' Personal Demographics

Table 1 describes the socio-economic characteristics of the respondents. The results indicated that 40.3% of the farmers are aged between 51 and 60 years, with an average age of 47.1 years. Nearly half of the respondents (50.3%) have an education attainment level less than high school completion

(50.3%) and only 17.2% have a bachelor's degree. With regard to farming experience, 49.7% farmers reported they have worked in the agriculture sector from 11 to 20 years, with an average of 15.06 years. More than half of the respondents (54%) revealed they do not access extension services. Most farms are characterized as small scale (47.7%), and the majority of respondents (64%) reported that they cultivated vegetables. Concerning soil fertility, the vast majority (88.9%) reported an average level of soil fertility on their farms, while only 3% reported highly fertile soil. Results also revealed that the overwhelming majority of respondents (91.9%) were not members of agricultural cooperatives, and that 86.3% have not accessed loans from agricultural banks.

**Table 1.** Demographic profile of respondents.

| Variable | Frequency | Percent |
|---|---|---|
| Age (*n* = 164) | | |
| 29–40 years | 52 | 31.7 |
| 41–50 years | 43 | 26.2 |
| 51–60 years | 66 | 40.3 |
| ≥61 years | 3 | 1.8 |
| Mean = 47.10; SD * = 9.468; Range = 41; Low = 29; High = 70 | | |
| Education level (*n* = 163) | | |
| Less than high school | 82 | 50.3 |
| High School | 28 | 17.2 |
| Diploma | 25 | 15.3 |
| Bachelor | 28 | 17.2 |
| Farming experience (*n* = 163) | | |
| <10 | 51 | 31.3 |
| 11–20 | 81 | 49.7 |
| 21–30 | 28 | 17.2 |
| ≥31 | 3 | 1.8 |
| Mean = 15.06; SD = 7.781; Range = 49; Low = 1; High = 50 | | |
| Farm size (*n* = 155) | | |
| <1 hectare | 74 | 47.7 |
| 1–3 hectares | 48 | 31 |
| >3 hectares | 33 | 21.3 |
| Access to extension services (*n* = 150) | | |
| Yes | 69 | 46.0 |
| No | 81 | 54.0 |
| Types of farming activities * (*n* = 164) | | |
| Vegetables | 103 | 64.0 |
| Fruits | 58 | 35.6 |
| Crops | 49 | 30.4 |
| Level of soil fertility (*n* = 162) | | |
| Low | 13 | 8.0 |
| Average | 144 | 88.9 |
| High | 5 | 3.1 |
| Membership of cooperatives (*n* = 163) | | |
| Yes | 13 | 8.1 |
| No | 147 | 91.9 |
| Access to loans (*n* = 161) | | |
| Yes | 20 | 12.4 |
| No | 141 | 87.6 |

* More than one answer was allowed; percentages of categories do not add up to 100. SD = Standard Deviation

## 3.2. Farmers' Beliefs about Climate Change

Farmers' beliefs regarding climate change are presented in Table 2. The results demonstrate that the majority of farmers agree or strongly agree that variable or unusual weather is caused by human activities (89.6%), natural change (93.3%), or natural change and human activities combined (81%). Meanwhile, 42% of respondents believed that there is no evidence that climate change is occurring,

and 54.6% believed that such evidence exists. Furthermore, 18% of the farmers believed that climate change is not occurring.

**Table 2.** Distribution of farmers' beliefs regarding climate change.

| Item | n | Strongly Disagree % | Disagree % | Neutral % | Agreed % | Strongly Agreed % | Mean | SD |
|---|---|---|---|---|---|---|---|---|
| Climate change is occurring because of human activities | 163 | 3.7 | 0.6 | 6.1 | 47.9 | 41.7 | 4.23 | 0.88 |
| Climate change is occurring because of natural change | 163 | 3.1 | 1.2 | 2.5 | 54 | 39.3 | 4.25 | 0.82 |
| Climate change is occurring equally because of natural changes and human causes | 163 | 7.3 | 7.4 | 4.3 | 50.3 | 30.7 | 3.9 | 1.14 |
| Insufficient evidence that climate change is occurring * | 163 | 29.4 | 25.2 | 4.3 | 19.6 | 21.5 | 2.79 | 1.56 |
| Climate change is not occurring * | 163 | 46.6 | 31.9 | 4.3 | 12.9 | 4.3 | 1.96 | 1.19 |

Summated Mean = 19.63; SD = 3.10; Range = 12; Low = 13; and High = 25; Item Mean = 3.92. * Scores were reversed for the negative items.

Farmers are classified into three categories such as high, moderate and low based on their level of beliefs (Figure 2). Farmers who have high beliefs are confident in the truth or existence of climate change and attributed it to human activities and natural change. In other words, they are likely to adapt to climate change. Conversely, farmers with a low level of beliefs perceive that the climate is not changing and may not act in the same sense. Findings, as illustrated in Figure 2, show that the majority of farmers (65.6%) have moderate beliefs regarding the occurrence of climate change. Less than one-third of the respondents (30.7%) hold strong beliefs and only 1.7% hold low beliefs about climate change. This result is supported by the findings of Al-Mutairi et al. [13], who confirmed that 82% of the respondents in Tabouk region in North Saudi Arabia had a moderate level of beliefs regarding climate change. Furthermore, the results of Arbuckle et al. [14] showed that many American farmers believed that climate change is occurring and believe that adaptation strategies should be implemented. Ricart et al. [28] reported similar results in their analysis about the perception of climate change among farmers and the public in the EU. They conducted a desk review of different sources (literature, research projects, and public opinion services) over the period from 2008 to 2017. The study confirmed that climate change is interdependent with the belief that climate change is happening.

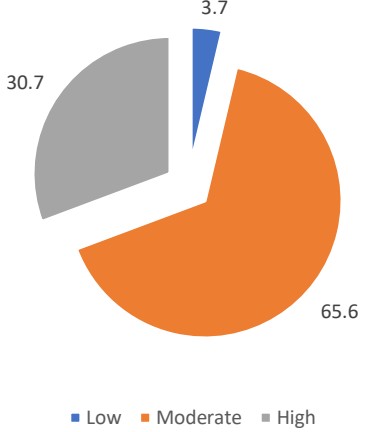

**Figure 2.** Level of farmers' beliefs about climate change.

To better understand farmers' beliefs, PAF was used to group items of belief into specific factors, as presented in Table 3. Two factors were extracted, explaining 49.53% of the total variance. Based on their factor loading values, Factor 1 comprised three items: (1) climate change is occurring because of human activity, (2) climate change is occurring because of natural change, and (3) climate change is occurring because of both human and natural changes. In contrast, Factor 2 included two items: (1) there is a lack of evidence that climate change is occurring and (2) climate change is not occurring. Items grouped under Factor 2 were selected for further analysis to group respondents based on their beliefs regarding the occurrence of climate change using K-mean cluster analysis (Table 4). Two clusters were identified, namely, Cluster 1 comprising 128 respondents and Cluster 2 comprising 35 respondents. The independent sample *t*-test results between the two clusters confirmed that Cluster 2 respondents believe less in the occurrence of climate change than Cluster 1 respondents. In other words, farmers in Cluster 1 believe in the occurrence of climate change.

**Table 3.** Principal axis factoring results.

| Climate Change Statements | Factor Loading Values | |
|---|---|---|
| | **Factor 1** | **Factor 2** |
| Climate change is occurring because of human activities | 0.8 | |
| Climate change is occurring because of natural change | 0.8 | |
| Insufficient evidence that climate change is occurring | | 0.77 |
| Climate change is occurring because of human and natural causes | 0.5 | |
| Climate change is not occurring | | 0.57 |

Factor 1: Percentage of explained variance = 31.98%; Factor 2: Percentage of explained variance = 17.55%.

**Table 4.** K-mean cluster analysis results.

| Climate Change Statements | Mean (SD) | | Mean Differences | *t* | *p* |
|---|---|---|---|---|---|
| | **Cluster 1 (*n* = 128)** | **Cluster 2 (*n* = 35)** | | | |
| Insufficient evidence about climate change is occurring | 2.29 (1.38) | 4.6 (0.5) | −2.31 | −9.71 | 0.00 |
| Climate change is not occurring | 1.48 (0.68) | 3.71 (1.02) | −2.23 | −15.37 | 0.00 |

### 3.3. Farmers' Concerns Regarding Climate Change

Farmers' concerns about the adverse impacts of climate change that they noticed on their own farms are listed in Table 5. Results indicated that farmers in the study area are moderately concerned about the harmful effects of climate change (summed score = 3.58). Pests, diseases, floods, and drought were the most observed impacts, with means of 3.31, 3.15, 3.15, and 3.11, respectively. These results are in line with the findings of Mase et al. [29], who argued that American farmers were most concerned with the impact of drought and increased heat stress on crops. Moreover, Orduño et al. [30], in their analysis about the most frequent weather patterns in Mexico, found that farmers were concerned about floods, hail, diseases, pests, and weed growth incidence.

Grouping farmers into three categories, as shown in Figure 3, indicates that farmers who have a high level of concern regarding climate change have greater interest in the impacts of climate change on their farming systems and livelihood. On the contrary, less concerned farmers are not worried about potential problems on their farms due to climate change. This attitude may affect their behavior toward implementing both adaptive and mitigative management strategies. As Figure 3 shows, most farmers (54.3%) are moderately concerned about climate change and almost a quarter (24.1%) have a high level of concern. In addition, only 21.6% of the farmers had low concern, indicating that the problems they encountered on their farms are believed to be the result of climate variability. In the Saudi Arabian context, Al-Mutairi et al. [13] established that 74% of respondents had moderate levels of concern regarding the impacts of climate change. In USA, Grimberg et al. [31] found that 48.3% of

farmers reported being concerned about climate change, while 86.7% reported being somewhat to very concerned about the impacts of climate change on agricultural production.

**Table 5.** Distribution of farmers' concerns about potential problems on their farms due to climate change.

| Item | n | Not Concerned % | Slightly Concerned % | Concerned % | Very Concerned % | Mean | SD |
|---|---|---|---|---|---|---|---|
| Increased drought | 162 | 1.2 | 32.1 | 43.8 | 22.8 | 3.11 | 1.13 |
| Increased flooding | 162 | 13.6 | 19.1 | 36.4 | 30.9 | 3.15 | 1.39 |
| Increased appearance of weeds | 162 | 6.2 | 21 | 53.1 | 19.8 | 3.06 | 1.11 |
| Increased insect pressure | 162 | 2.5 | 22.8 | 45.1 | 29.6 | 3.31 | 1.19 |
| Higher incidence of crops disease | 162 | 3.7 | 23.5 | 50 | 22.8 | 3.15 | 1.13 |
| Increased soil erosion | 162 | 9.3 | 25.9 | 46.9 | 17.9 | 2.91 | 1.16 |
| Increased heat stress on crops | 162 | 2.5 | 27.2 | 52.5 | 17.9 | 3.04 | 1.04 |
| Increased saturated soils and ponded water | 162 | 4.3 | 25.9 | 56.2 | 13.6 | 2.93 | 0.98 |

Summated Mean = 22.91; SD = 4.84; Range = 24; Low = 8; and High = 32; Item Mean = 2.86.

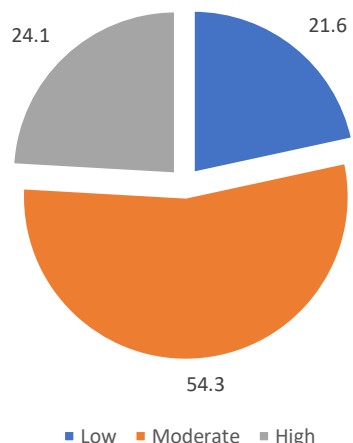

**Figure 3.** Level of farmers' concern about climate change.

*3.4. Differences in Farmers' Beliefs and Concerns according to Their Socio-Economic Characteristics*

3.4.1. Membership of Agricultural Cooperatives

Table 6 shows a significant difference in beliefs regarding climate change between members and non-members of agricultural cooperatives ($p < 0.05$). The effect size of the difference between the summed means of these two groups was moderate (Cohen's $d = 0.65$). This means that farmers who are members of agricultural cooperatives are less likely to believe in the occurrence of climate change (mean = 17.50) than non-members (mean = 19.82). Despite the importance of agricultural cooperatives in enabling members to manage climate risks and adapt to climate change [32], the membership of an agricultural cooperative is not an influencing factor on increasing farmers' beliefs. This is partly due to a weak role of agricultural cooperatives in enhancing farmers' perception about farming problems, as indicated during the interviews. This role affects the vast majority of the respondents in joining the cooperatives. Individual farmers deal with different stakeholders in the agricultural value chain to

purchase inputs and for crop marketing more than farmers who belonged to cooperatives. This tendency may facilitate non-members to access more information about climate change and exchange views with others. The finding is consistent with the results achieved by Ado et al. [33], who found that membership of agricultural cooperatives does not significantly affect farmers' awareness of climate change. With regard to farmers' concerns, Table 6 shows no significant difference between members and non-members of agricultural cooperatives ($p > 0.05$).

**Table 6.** The *t*-test comparison for differences in farmers' beliefs and concerns with regard to socio-economic characteristics.

| Variables | Farmers' Beliefs | | | | Farmers' Concerns | | | |
|---|---|---|---|---|---|---|---|---|
| | Mean | SD | *t* | Sig 2-Tail | Mean | SD | *t* | Sig 2-Tail |
| | Membership of agricultural cooperatives | | | | | | | |
| Yes, *n* = 13 | 17.50 | 3.09 | −2.67 | 0.008 Cohen's *d* = 0.65 | 25.20 | 4.03 | 1.74 | 0.084 |
| No, *n* = 147 | 19.82 | 3.04 | | | 22.71 | 4.91 | | |
| | Access to loans | | | | | | | |
| Yes, *n* = 20 | 18.30 | 2.99 | −2.06 | 0.014 Cohen's *d* = 0.49 | 26.60 | 4.04 | 3.74 | 0.000 Cohen's *d* = 0.94 |
| No, *n* = 141 | 19.81 | 3.08 | | | 22.42 | 4.75 | | |
| | Access to extension services | | | | | | | |
| Yes, *n* = 69 | 20.75 | 3.02 | 6.35 | 0.000 Cohen's *d* = 0.98 | 22.45 | 4.76 | 1.58 | 0.116 |
| No, *n* = 80 | 17.85 | 2.55 | | | 23.68 | 4.71 | | |

### 3.4.2. Access to Loans

Access to loans represented a significant difference in farmers' beliefs at the 0.01 level (Table 6). Farmers who have not received loans from agricultural banks have higher level climate change beliefs (mean = 19.81) compared to farmers who have received loans (mean = 18.30). Based on the summed score, the differences in the mean represent a moderate effect (Cohen's *d* = 0.49). Access to credit plays a critical role in enhancing farmers' perception of climatic changes and transforming agribusiness models to be more climate-smart and inclusive for small farmers [34]. During the field study, we noticed that some farmers who have not received loans adopted climate-smart agriculture (CSA) practices. Farmers who establish these investments often are motivated not only by direct costs and returns but also by increased resilience, a higher degree of certainty regarding future revenue streams, and the prospect of reduced volatility [35]. The findings in Table 6 also show significant differences in farmers' concerns, according to whether or not they received loans from agricultural banks ($t = 3.74$; $p < 0.05$). Regarding the size of differences between the two groups, the result of Cohen's *d* = 0.94 showed a high level of difference, meaning that farmers who accessed loans have a significantly higher level of concern in relation to climate change (mean = 26.60) than those who did not access loans (mean = 22.42). This is supported by the findings of research by Fosu-Mensah et al. [36] and Debela et al. [37], that indicated that increased access to agricultural loans will further enhance farmers' concerns about climate change and result in better management of climate-induced risks.

### 3.4.3. Access to Extension Services

Table 6 shows that farmers who have not accessed extension services have significantly higher beliefs in climate change compared to those who have accessed to extension services ($t = 6.35$; $p < 0.05$), and the difference in the mean represents a significant effect (Cohen's *d* = 0.98). It is surprising to find that the use of extension services does not affect farmers' beliefs in climate change. In general, farmers who have access to extension services will have more awareness; however, there is no evidence that, in Saudi Arabia during the past five years, extension programs about climate change and adaptation have been delivered to farmers. Therefore, it is certain that extension will not have an effect due to the lack of both knowledge among farmers and extension agents. This result corresponds to the findings of previous research [36,38,39].

### 3.4.4. Level of Education

Results of the one-way ANOVA (Table 7) revealed no significant differences in farmers' beliefs ($F = 0.93$; $p > 0.05$) or concerns ($F = 0.44$; $p > 0.05$) based on the level of educational attainment.

### 3.4.5. Types of Farming Activities

Table 7 shows that that there are no significant differences in farmers' beliefs or concerns based on the type of farming activities at the 0.05 level.

### 3.4.6. Soil Fertility

Table 7 shows the results of the one-way ANOVA. Results indicated significant differences in farmers' beliefs regarding climate change depending on soil fertility ($F = 4.21$; $p < 0.05$). The size difference between the groups was low (partial eta squared = 0.05). The Games–Howell post hoc test indicated that farmers who own farms with average soil fertility (mean = 19.85) have a significantly higher mean as compared to other groups. This confirms that soil fertility level is a significant determinant of farmers' beliefs regarding climate change. Soil fertility is an important factor in climate change adaptation. According to Stucker and López-Gunn [40], soil fertility, water availability and timing, and resilience to natural disasters are the most important factors in determining whether crop yields increase or decrease in any given year. Such factors are the main determinants of both livelihood security and environmental health. This finding is supported by the work of both Fosu-Mensah et al. [36] and Huong et al. [38], who found that farmers with fertile soil are more likely to adopt climate change adaptation practices. However, no significant differences were found in farmers' concerns based on levels of soil fertility ($F = 2.58$; $p > 0.05$).

### 3.4.7. Age

Table 7 demonstrates that farmers' age had a significant effect on their beliefs about climate change ($F = 5.22$; $p > 0.01$). The size differences between means of the groups was small (partial eta squared = 0.090). The group age of 51 to 60 years had a significantly higher mean (mean = 20.52) than other groups, according to the results of the Games–Howell post hoc test. This means that older farmers in the study area indicated higher levels of beliefs regarding climate change. As noted by Bonem et al. [41], older farmers had more farming experience and rated themselves as more likely to observe the problems or changes over time than young farmers. This result is in agreement with the findings of Shrestha and Baral's [42] research, but is inconsistent with the findings of some other research [38,43]. In contrast, no significant differences were found regarding farmers' concerns.

### 3.4.8. Farm Size

Results showed that farm size is considered a determinant of beliefs about climate change, and it has a significant effect on the farmers' beliefs at the $p < 0.01$ level of significance (Table 7). Specifically, larger farm size is associated with higher levels of belief about climate change. A larger farm size generally implies greater overall economic losses if crops are damaged, forcing larger-scale farmers to acquire information about climate change to mitigate potential risks [18]. This suggests that a larger farm size enhances farmers' beliefs about climate variability, while the risk of climate change to smaller farms may increase. This finding is supported by Mustafa et al. [39], who discovered that, in Pakistan, farmers' levels of belief were determined by farm size. However, the results revealed no significant differences in farmers' concerns regarding farm size.

**Table 7.** ANOVA differences in farmers' beliefs and concerns about climate change with regard to socio-economic characteristics.

| Variables | Farmers' Beliefs | | | | Farmers' Concerns | | | |
|---|---|---|---|---|---|---|---|---|
| | Mean | SD | F | Sig 2-Tail | Mean | SD | F | Sig 2-Tail |
| Level of education | | | | | | | | |
| Less than high school *n* = 82 | 19.60 | 3.01 | | | 23.26 | 5.25 | | |
| High school *n* = 28 | 20.25 | 3.19 | 0.938 | 0.424 | 23.07 | 4.46 | 0.442 | 0.723 |
| Diploma *n* = 25 | 18.84 | 3.00 | | | 22.76 | 4.00 | | |
| Bachelor *n* = 27 | 19.81 | 3.43 | | | 22.03 | 4.62 | | |
| Types of farming activity | | | | | | | | |
| Vegetables *n* = 102 | 19.32 | 3.00 | | | 22.66 | 4.77 | | |
| Fruits *n* = 58 | 20.55 | 2.78 | 1.42 | 0.243 | 21.00 | 4.44 | 1.82 | 0.164 |
| Crops *n* = 49 | 20.08 | 3.31 | | | 23.89 | 5.00 | | |
| Level of soil fertility | | | | | | | | |
| Low *n* = 13 | 18.07 | 3.42 | | | 25.23 | 3.91 | | |
| Average *n* = 144 | 19.85 | 3.01 | 4.21 | 0.016 | 22.82 | 4.86 | 2.58 | 0.079 |
| High *n* = 5 | 16.80 | 2.68 | | | 19.80 | 5.31 | | |
| Age | | | | | | | | |
| 29–40 years *n* = 51 | 18.50 | 3.04 | | | 22.76 | 4.03 | | |
| 41–50 years *n* = 43 | 19.81 | 2.91 | 5.22 | 0.002 | 22.77 | 4.36 | 0.704 | 0.551 |
| 51–60 years *n* = 66 | 20.52 | 2.93 | | | 23.30 | 5.47 | | |
| ≥61 years *n* = 3 | 17.00 | 4.58 | | | 19.34 | 9.86 | | |
| Farm size | | | | | | | | |
| <1 hectare *n* = 74 | 17.66 | 1.05 | | | 23.41 | 4.45 | | |
| 1–3 hectares *n* = 48 | 18.92 | 2.75 | 4.74 | 0.008 | 22.98 | 3.66 | 0.51 | 0.48 |
| >3 hectares *n* = 33 | 20.44 | 2.33 | | | 23.56 | 4.11 | | |

### 3.5. Farmers' Climate Change Capacity Building

Farmers' perceptions of the role of agricultural extension in building their climate change adaptive capacity were examined for the two clusters of beliefs (Table 8). Results revealed that farmers in Cluster 1 reported higher means compared to those in Cluster 2. In other words, farmers who are more likely to believe in climate change are relatively more in agreement with the role of extension services in capacity building than those with less belief in the occurrence of climate change. However, independent sample *t*-test results showed a statistically significant difference between the two clusters for three statements: farmers' food storage, processing, and utilization training ($t = 2.64$; $p = 0.009$); awareness creation and building capacity of extension staff ($t = 3.17$; $p = 0.002$); and use of information communication technologies to create awareness ($t = 2.91$; $p = 0.004$). These results reflect the crucial role that extension services continue to play in dealing with farming challenges, including cross-cutting issues such as climate change. Kalimba and Culas [44] suggested that governments should invest in extension services to facilitate information dissemination, change farmers' attitudes, implement training programs, and network with other stakeholders interested in climate change issues to strengthen extension services to meet the specific needs of farmers.

**Table 8.** Differences between clusters of farmers according to their perceptions of the role of extension services in building capacity.

| Capacity Building Statements | Mean (SD) | | Mean Differences | *t* | *p* |
| --- | --- | --- | --- | --- | --- |
| | Cluster 1 ($n = 128$) | Cluster 2 ($n = 35$) | | | |
| Conduct awareness meetings with farmers to sensitize them to climate change. | 4.02 (0.78) | 3.80 (1.08) | 0.22 | 1.32 | 0.188 |
| Conduct field days to publicize new and improved drought and disease resistant technologies for crops, livestock. | 4.13 (0.73) | 4.00 (1.00) | 0.13 | 0.82 | 0.412 |
| Conduct demonstrations to provide farmers with new knowledge and skills related to climate change adaptation technologies. | 4.09 (0.86) | 3.91 (0.95) | 0.18 | 1.07 | 0.285 |
| Use farmer-to-farmer extension methods to promote awareness and adoption of climate change adaptation best practices. | 4.21 (0.71) | 3.94 (1.08) | 0.27 | 1.76 | 0.081 |
| Train farmers in food storage, processing, and utilization methods to increase food shelf life and reduce postharvest losses. | 4.16 (0.73) | 3.77 (0.94) | 0.39 | 2.64 | 0.009 * |
| Disseminate information on weather focus and early warnings to allow for better planning. | 4.12 (0.82) | 3.91 (1.04) | 0.20 | 1.22 | 0.223 |
| Use farmer field schools to train farmers in available adaptation options to suit local conditions. | 3.95 (0.86) | 3.83 (0.99) | 0.12 | 0.69 | 0.493 |
| Link small-holder farmers to agricultural research institutions for on-farm adaptive research on climate change adaptation best practices in a variety of farming systems. | 4.27 (0.75) | 4.00 (1.11) | 0.27 | 1.71 | 0.09 |
| Build capacity and create awareness among extension staff so they have knowledge and skills to promote adaptation interventions. | 4.34 (0.66) | 3.89 (1.05) | 0.46 | 3.17 | 0.002 * |
| Use information communication technologies, such as radio and cell phones, to create awareness among farmers of climate change issues and adaptation options. | 4.36 (0.71) | 3.89 (1.23) | 0.47 | 2.91 | 0.004 * |

Note: * $p < 0.01$.

## 4. Conclusions

This study provided insights into Saudi Arabian farmers' beliefs and concerns regarding climate change. Results indicated that most farmers in the study area believed that climate change is caused by human activities and natural change. Furthermore, the majority of farmers are concerned about insects and disease pressures, flooding, and an increase in heat stress on crops. Although the findings emphasize the weakness of extension services in increasing the awareness of climate change, the respondents who are more likely to believe in climate change argued that extension could play a significant future role in capacity building. The study provides useful implications for policy makers. Implementing innovative extension approaches for climate-smart agriculture are crucial to encourage farmers and rural communities to adopt suitable adaptation strategies. Therefore, there is a need for reviewing the best practices adopted by the other countries to strengthen the contribution of extension in combating climate change. One of the successful stories is the implementation of the Farmer Field Schools (FFS) approach. FFS promote social learning among farmers by developing problem-solving skills, increasing their technical knowledge, and enhancing their decision-making abilities. This in turn has potential to have a major impact on farmers' understanding of the different components of an agroecosystem. Establishing plant clinics is another innovative approach. Plant clinics provide diagnosis of and recommendations for any problem and any crop, and snapshots of problems on farmers' fields are fed into a global knowledge bank. The outcome of this approach is to increase the farming system efficiency by reducing crop losses. This contributes to climate change mitigation by reducing both direct and indirect emissions. In addition, awareness campaigns using information technologies, such as social media, should be designed to improve farmers' access to climate change information and, therefore, enhance farmers' levels of adaptation. Future research should examine the relationship between farmers' beliefs and concerns regarding the impact of climate change, as well as farmers' risk perceptions, decision-making processes, and the factors that influence their beliefs, concerns, attitudes, and behaviors relating to climate change. This study was limited to Abu Arish governorate of Jazan region. The results may not be generalizable to farmers who live in different geographic regions. Therefore, it is recommended that similar research be conducted in other parts of Saudi Arabia that are more prone to climate change disasters.

**Author Contributions:** Conceptualization, methodology, writing—original draft preparation B.A.A.; Writing—review and editing H.S.K.; data analysis R.K.N. and data curation M.M. All authors have read and agreed to the published version of the manuscript.

**Funding:** The authors would like to extend their sincere appreciation to the Deanship of Scientific Research, King Saud University, Saudi Arabia, for funding this research through Research Group No. RGP—1440-016.

**Acknowledgments:** The authors are grateful to the Deanship of Scientific Research and RSSU at the King Saud University for their technical support.

**Conflicts of Interest:** The authors declare no conflict of interest.

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
