# Peer review of "Farmers’ Beliefs and Concerns about Climate Change: An Assessment from Southern Saudi Arabia"

_agriculture, doi:10.3390/agriculture10070253_

Round 1
Reviewer 1 Report
Thank you for a well written, interesting and informative manuscript. I have made a number of minor comments in the attached file that should, once addressed, make the manuscript easier to follow.
I would note that the conclusions are good, but some of the implications and detail could be strengthened and made more specific. I would try to use more concrete language than 'could' or 'should'. While some examples of extension services of policies are given, these could be expanded with a bit more detail or additional examples.
The English language is of high quality, however, there are a few awkward wordings/phrasings. It would be worth having a native English speaker who is not all that familiar with the work go through it for a final language check before publication.

Author Response
Point 1: Aren't most farming systems reliant on the climate? I think either remove this sentence or add to it, i.e. "... highly reliant on favorable climatic conditions." where you then define favorable (the sentence starting at line 46).
Response 1: We modified this sentence in the introduction.
Point 2: While I know that downscaled model predictions have a relatively high level of uncertainty, are there any indications of what climatic shifts might be seen in the Jazan region? I think this could be useful context to your case study region. As you extrapolate your findings from the interviews in Jazan to the rest of the country, this could provide more 'credibility' if the expected conditions are similar to the rest of the country.
Response 2: We clarified the climatic pattern of the study area (Jazan region) in the methodology section. Jazan is only similar to other regions in the south of Saudi arabia. Saudi Arabia has three different agro climatic zones; North, Middle, and South. The generalization of the findings could be only suitable for the Southern areas of KSA.
Point 3: While extension programmers and awareness campaigns are interesting, it would be even more interesting to understand if transformational shifts to entirely new products or farming systems might occur under climate change.
Response 3: We modified this sentence in the introduction.
Point 4: This clause doesn't really fit anywhere there. I would remove it unless you are going to add in other details of the administrative/governance arrangements of the region.
Response 4: We added other details to this sentence in the methodology section.
Point 5: This paragraph doesn't flow very well. It jumps from the location and area and number of farms to rainfall back to size of farms to production and back to water/irrigation and soil. My suggestion is to rearrange the paragraph as follows: Location and area (current first 2 sentences + fertile loamy soil) -> Size of farms and production -> variable rainfall and temperature, so water for crops is sourced from... and drip irrigation.
Response 5: We rearranged the paragraph based this comment (highlighted).
Point 6: Map needs to be much higher resolution. It is quite difficult to read.
Response 6: We designed a new map.
Point 7: It isn't clear - was data collected through a survey (more details needed if online/ paper-based, response rates) and interviews or if it may be a terminology issue that the survey instrument was the interview questions. Or, was there a series of interviews in addition to a survey? Please clarify this.
Response 7: We addressed the comments. We added a few sentences (highlighted).
Point 8: More details about the interviews. Were they semi-structured or structured? Was there any accounting for differences in the way that interviewers may have conducted the interviews?
Response 8: We used semi-structured interviews to collect field data. This point added to the methodology (highlighted).
Point 9: Could you add in a bit more explanation as to the meaning of what moderate, strong and weak beliefs about the occurrence of climate change? Is it that they might possibly thing that climate change might be happening, the climate is really changing, and the climate isn't changing, respectively?
Response 9: We explained all the categories of beliefs and concerns in the results section (highlighted).
Point 10: The differences in beliefs according to social and economic characteristics could benefit from some more discussion and/or qualitative material, if available. Some of the questions that spring to mind are "why?" Why does membership in a cooperative mean that they are less likely to believe in climate change, but only moderately? What is it about loans that influences their beliefs? The same question could be asked about most of the sections here.
Response 10: We added more discussion to interpret the results of each variable
(highlighted).
Point 11: I find this quite interesting. I hope that you address it further in the discussion. Did you do any work on whether the size of the loan influenced their beliefs around climate change? (These questions are out of interest; I am not looking for you to re-do your research in order to answer these questions. They may make up part of a follow-up research project.
Response 11: Thank you for your valuable comment. We will reconsider this result in future research. More explanations were added to this point in the results (highlighted).
Point 12: Both statements here are "have not accessed extension services" Is it that those that have accessed them have higher belief in climate change? (This is what comes across in the conclusions)
Response 12: We modified this sentence (highlighted).
Point 13: This may be more of a comment for the editorial staff, but it seems to me to make more sense to have table 6 here, rather than having to scroll down a couple of pages to reach it.
Response13: We rearranged tables 5 and 6 according to this comment.
Point 14: Conclusion. This doesn't come through in the results, primarily, I think, due to a typo. Please see my comment earlier.
Response 14: We modified this sentence in the conclusion to match the results.
Point 15: I would note that the conclusions are good, but some of the implications and detail could be strengthened and made more specific. I would try to use more concrete language than 'could' or 'should'. While some examples of extension services of policies are given, these could be expanded with a bit more detail or additional examples.
Response 15: We added more examples for extension services that could be implemented in the conclusion section (highlighted).

Reviewer 2 Report
Dear Authors,
The research carried out is interesting. I think you should better explain why it was done. The way of conducting the research, the methods used are correct I miss the better described application of these research.
The authors in Conclusions write: “Future research should examine the relationship between farmers’ beliefs and concerns regarding the impact of climate change as well as farmers’ risk perceptions, decision-making processes, and the factors that influence their beliefs, concerns, attitudes, and behaviors relating to climate change.” It seems that there should be some more specific solutions here.
I think the article will be more complete if it is added:
- a fuller explanation of why these studies were carried out
- propose a solution or indicate what may happen if no action is taken.
Technical Notes:
- 1 is not readable
- I would consider duplicating the % values in the text and table. The data from the tables are duplicated in the text, I would consider it necessary. Can you first make a short introduction to the table, put the table in and describe it?
Have the authors wondered how their research fits into Sustainably Development Goals?
Author Response
Point 1: The research carried out is interesting. I think you should better explain why it was done. The way of conducting the research, the methods used are correct I miss the better described application of these research. (a full explanation of why these studies were carried out).
Response 1: We added a paragraph in the introduction about why it was done (highlighted).
Point 2: The authors in Conclusions write: “Future research should examine the relationship between farmers’ beliefs and concerns regarding the impact of climate change as well as farmers’ risk perceptions, decision-making processes, and the factors that influence their beliefs, concerns, attitudes, and behaviors relating to climate change.” It seems that there should be some more specific solutions here. (propose a solution or indicate what may happen if no action is taken).
Response 2: We added more specific solutions that could be implemented in the conclusion section (highlighted).
Point 3: figure 1 is not readable
Response 3: We designed a new map.
Point 4. Can you first make a short introduction to the table, put the table in and describe it?
Response 4. We addressed this comment along the results.
Point 5: Have the authors wondered how their research fits into Sustainably Development Goals?
Response 5. We added new sentences in the beginning of the introduction to illustrate this point. Also, it highlighted at the importance of the research at the end of the introduction section.
